# Inhibitory Effect of Quercetin on Oxidative Endogen Enzymes: A Focus on Putative Binding Modes

**DOI:** 10.3390/ijms242015391

**Published:** 2023-10-20

**Authors:** Stefania Olla, Chiara Siguri, Antonella Fais, Benedetta Era, Massimo Claudio Fantini, Amalia Di Petrillo

**Affiliations:** 1Istituto di Ricerca Genetica e Biomedica, Consiglio Nazionale delle Ricerche, Cittadella Universitaria di Monserrato, 09042 Monserrato, Italy; chiara.siguri@irgb.cnr.it; 2Department of Life and Environmental Sciences, University of Cagliari, 09042 Monserrato, Italy; fais@unica.it (A.F.); era@unica.it (B.E.); 3Department of Medical Science and Public Health, University of Cagliari, 09042 Monserrato, Italy; massimoc.fantini@unica.it

**Keywords:** quercetin, xanthine oxidase, myeloperoxidase, nicotinamide adenine dinucleotide phosphate oxidase, lipoxygenase, monoamine oxidase, molecular docking

## Abstract

Oxidative stress is defined as an imbalance between the production of free radicals and reactive oxygen species (ROS) and the ability of the body to neutralize them by anti-oxidant defense systems. Cells can produce ROS during physiological processes, but excessive ROS can lead to non-specific and irreversible damage to biological molecules, such as DNA, lipids, and proteins. Mitochondria mainly produce endogenous ROS during both physiological and pathological conditions. Enzymes like nicotinamide adenine dinucleotide phosphate oxidase (NOX), xanthine oxidase (XO), lipoxygenase (LOX), myeloperoxidase (MPO), and monoamine oxidase (MAO) contribute to this process. The body has enzymatic and non-enzymatic defense systems to neutralize ROS. The intake of bioactive phenols, like quercetin (Que), can protect against pro-oxidative damage by quenching ROS through a non-enzymatic system. In this study, we evaluate the ability of Que to target endogenous oxidant enzymes involved in ROS production and explore the mechanisms of action underlying its anti-oxidant properties. Que can act as a free radical scavenger by donating electrons through the negative charges in its phenolic and ketone groups. Additionally, it can effectively inhibit the activity of several endogenous oxidative enzymes by binding them with high affinity and specificity. Que had the best molecular docking results with XO, followed by MAO-A, 5-LOX, NOX, and MPO. Que’s binding to these enzymes was confirmed by subsequent molecular dynamics, revealing different stability phases depending on the enzyme bound. The 500 ns simulation showed a net evolution of binding for NOX and MPO. These findings suggest that Que has potential as a natural therapy for diseases related to oxidative stress.

## 1. Introduction

Quercetin (Que) is a 3,3′,4′,5,7-pentahydroxyflavone with the chemical formula C_15_H_10_O_7_. The presence of five hydroxyl groups in the structure and the pyrocatechol-kind of benzene ring make Que a strong anti-oxidant and a good scavenger of free radicals [1]. The structure of Que contains a keto carbonyl group, and the oxygen atom present on the first carbon, being basic, can form salts with acids. Furthermore, a dihydroxy group between the A ring, o-dihydroxy group B, C ring C2, C3 double bond, and 4-carbonyl are the active groups in Que (Appendix A). Que’s biological activities have been largely attributed to these active phenolic hydroxyl groups and double bonds [2].

Que is highly present in daily human foods, such as onions, apples, red wine, and tea. It has various biological effects, including anti-oxidant, anti-cancer, anti-inflammatory, anti-viral, and anti-aging effects. Additionally, it has anti-aggregant and vasodilator effects, which help protect against cardiovascular diseases [3,4]. In humans consuming flavonoid-rich foods, the concentration range of Que plasma levels is 0.3–7.6 μM, mainly in the form of glucuronidated and sulfated metabolites [5].

Free radicals and reactive oxygen species (ROS) are produced in cells due to common metabolic processes or external sources [6]. Excessive oxidative stress disrupts the balance between the oxidation and anti-oxidation systems. This causes non-specific and irreversible damage to biological molecules, such as lipids, proteins, and DNA, leading to a loss of function [7].

In many chronic diseases, such as cancer, neurodegenerative diseases, and metabolic diseases, oxidative stress is often the primary trigger [8].

Possible sources of endogenous ROS include the nicotinamide adenine dinucleotide phosphate oxidase (NOX), xanthine oxidase (XO), lipoxygenase (LOX), myeloperoxidase (MPO), and monoamine-oxidase (MAO) enzymes [9]. Increased activation of these enzymes has been linked to oxidative stress as well as the onset and advancement of inflammatory diseases.

Notably, Que has exhibited promising protective effects by attenuating the expression and enzymatic activity of these endogenous oxidative enzymes [10,11,12,13,14,15].

NOX (EC 1.6.3.1) is a transmembrane glycoprotein that catalyzes the production of superoxide (O_2_^−^) from oxygen and NADPH. O_2_^−^ reacts quickly to produce a burst of additional oxidants, including hydrogen peroxide (H_2_O_2_), which is typically further converted by MPO into hypochlorous acid (HOCl) and hydroxyl radical (•OH). For this reason, NOX is usually considered a major source of ROS and oxidative stress in eukaryotic cells [16].

XO (EC 1.17.3.2) is an iron-molybdenum flavoprotein, predominantly found in the cytoplasm of mammalian tissues, that plays a crucial role in purine catabolism [17]. It catalyzes the oxidation of hypoxanthine to xanthine and xanthine to uric acid (UA) with the production of a superoxide anion. The accumulation of UA has been shown to initiate the inflammatory process through the NLRP3 inflammasome and ROS production, contributing to inflammation-related tissue damage [18].

MPO (EC 1.11.2.2) is an enzyme found in the azurophilic/primary granules of neutrophils and, to a lesser extent, in monocytes. MPO catalyzes the oxidation of chloride and other halide ions in H_2_O_2_ to generate HOCl and other highly reactive products that mediate efficient anti-microbial action [19].

LOXs (EC 1.13.11.12) are a large monomeric protein family with non-heme, non-sulfur, iron cofactor-containing dioxygenases that catalyze the oxidation of polyunsaturated fatty acids (PUFA) in lipids containing a cis,cis-1,4-pentadiene into cell signaling agents [20]. The main products of LOXs are leukotrienes and lipoxins, which play important roles in several inflammation-related diseases, such as arthritis, asthma, cancer, and allergies.

MAO (EC 1.4.3.4) is a riboflavin protein distributed on the outer mitochondrial membrane that catalyzes the oxidative deamination of biogenic and xenobiotic amines, producing the corresponding aldehydes, hydrogen peroxide, and ammonia [21]. Two isoforms of MAO, MAO-A and MAO-B, differ in their substrate and tissue distributions. MAO-A preferentially deaminates serotonin and norepinephrine, while MAO-B metabolizes dopamine [22]. The generation of H_2_O_2_ via MAOs is reported to be a cytotoxic factor involved in oxidative stress and neurodegenerative disorders, such as Parkinson’s disease [23].

Due to their hydrophobic nature, Que and other phenolic compounds like curcumin exhibit a strong affinity for mitochondria and can be traced through autofluorescence. This unique property of Que may contribute to its specific effect on mitochondrial dysfunction [24].

It is also important to address the limitations of Que’s bioavailability, especially in its aglycone form, which has low solubility in water and gastrointestinal fluids, leading to precipitation and reduced absorption [25]. Various que derivatives have differing bioavailabilities, with que glycosides found in onions having the highest absorption rates. These challenges can be overcome through strategies like encapsulation in colloidal particles and crystal engineering, as demonstrated in recent articles [26,27].

Our comprehensive investigation delves into Que’s potential to target oxidant enzymes implicated in ROS production. To explore these aspects thoroughly, we conducted docking analyses and molecular dynamics (MD) simulations to validate the obtained data and calculate the MMGBSA binding free energy before and during the entire dynamics. The use of docking allowed us to explore binding interactions between Que and targets, providing insight into initial structural arrangements and potential binding modes. On the other hand, the MD simulations performed provided a dynamic view of the ligand-protein complex, allowing us to assess the stability of the binding and observe any conformational changes over time. By calculating the free energy of binding with MMGBSA, we quantified the thermodynamic aspects of the ligand-protein interaction.

## 2. Results

### 2.1. Quercetin Anti-Oxidant Activity: In Silico Approach

Density functional theoretical (DFT) approaches [28] at the B3LYP/6-311++G(d, p) level were applied for the optimization of Que, and revealed the presence of C2′H/C3OH, C3OH C4O, and C4O/C5O H bonds (Appendix A). Moreover, optimized geometry (Appendix A), electrostatic potential (ESP), average local ionization energy (ALIE), electron density (ED), highest occupied molecular orbital (HOMO), lowest un-occupied molecular orbital (LUMO), and natural bond orbitals (NBO) were calculated.

ESP analysis is a robust tool for revealing the chemical and reactivity attributes of molecular systems [29]. In our computations, the ESP values for Que range from −2.65 eV to 3.18 eV (Table 1).

These values provide insights into the distribution of electron-rich and electron-poor regions within the molecule, indicating potential sites of high and low electron density (Appendix A).

ALIE, a pivotal property measuring the energy required to extract an electron from a specific location within the molecular system, reveals a range for Que from 8.84 eV to 16.81 eV (Table 1). Lower ALIE values signify regions where electrons are less tightly bound, suggesting sites with higher reactivity (Appendix A).

The ED analysis involves determining the probability of locating an electron in the vicinity of a molecule [30]. This property contributes to understanding the electronic structure and behavior of Que. Our computations (Appendix A) provide valuable insights into the spatial distribution of electron density around the molecule.

The HOMO and LUMO orbitals were visually presented through color-coded surfaces (Figure 1). The HOMO and LUMO energy levels were determined to be −5.68 eV and −1.94 eV, respectively, and their energy difference, known as the energy gap (ΔE), was found to be 3.74 eV (Table 1). This gap indicates Que’s capacity for engaging in free radical optimization mechanisms, with smaller ΔE values suggesting a greater propensity for effective participation in radical scavenging reactions [31]. In accordance with Koopman’s theorem (1934), the LUMO energy level corresponds to the electron affinity (EA = −ELUMO), indicating its ability to accept electrons during molecular interactions, and the HOMO energy level is associated with the ionization potential (IP = −EHOMO), signifying its role as an electron donor. For Que, the IP was calculated as 5.68 eV and the EA as 1.94 eV (Table 1).

Natural bond orbital (NBO) analysis is a robust method for studying intra-molecular and inter-molecular bonding, charge transfer, and conjugative interactions within molecular systems. Higher stabilization energy values indicate stronger electron donor-acceptor interactions, especially enhancing conjugative interactions [32]. The results of NBO studies, including natural population analysis, can be found in Appendix A, while the second-order perturbation analysis of the Fock matrix is presented in Appendix A.

### 2.2. Molecular Docking

Molecular docking was performed for Que and all selected endogenous oxidative enzymes. Table 2 shows the information regarding the five X-rays used. For all docking results obtained, the binding free energy of the receptor-ligand complex was predicted using MMGBSA calculations [33].

#### 2.2.1. NAD(P)H Oxidase

The results of docking showed that Que establishes H-bonds with Asp 179 and Cys 242 and interacts with the different residues of NOX shown in Figure 2 with a Glide Gscore of −5.35 kcal/mol (Table 3). Subsequently, in this docking pose, the ligand-receptor binding energy was calculated, obtaining a MMGBSA dG bind of −4.82 kcal/mol (Table 3).

#### 2.2.2. Xanthine Oxidase

The results of docking on XO showed a Glide Gscore of −8.87 kcal/mol and that Que establishes 4 four H-bonds with Glu 802, Ser 876, Arg 880, and Thr 1010, respectively, and two π-π interactions with Phe 914 and Phe 1009, as well as interactions with the residues shown in Figure 3. The best pose resulting from the docking simulation was then superimposed on the native Que, showing an RMSD of 0.111 Å (Appendix A) and a similar orientation of the binding mode. As before, the docking results were subjected to MMGBSA calculations, obtaining a free energy of −22.18 kcal/mol (Table 3).

#### 2.2.3. Myeloperoxidase

The results of docking on MPO revealed a Glide Gscore of −4.91 kcal/mol and that Que is oriented within the active site of MPO with the A-ring and C-ring positioned over the iron heme, a crucial component of MPO catalytic activity. This orientation suggests a potential mechanism of inhibition of MPO activity by Que. Que interacts with Phe 147 and Phe 407 via π-π interactions and establishes two H-bonds with Glu 116 (Figure 4). Subsequently, the binding between Que and MPO obtained by docking was subjected to the MMGBSA calculation, obtaining a dG binding of −28.81 kcal/mol (Table 3).

#### 2.2.4. Lipoxygenase

The results indicated that Que establishes several crucial interactions with amino acid residues within the 5-LOX binding site and has a Glide Gscore of −5.94 kcal/mol. Specifically, Que formed five H bonds with His 130, Glu 134, Thr 137, Arg 138, and Asp 166, and engaged in π-cation interactions with Arg 101 (Figure 5). Moreover, the binding pose was subjected to the MMGBSA calculation, obtaining a dG binding value of −37.42 kcal/mol (Table 3).

#### 2.2.5. Monoamine Oxidase

The results indicated a Glide Gscore of −8.14 kcal/mol and that Que exhibited interactions with specific amino acid residues within the MAO-A active site. Notably, Que formed a H-bond with Ala 111 and engaged in a π-π interaction with Phe 208, as illustrated in Figure 6. Then, the obtained docking pose was subjected to the MMGBSA calculation with a positive free energy of 5.72 kcal/mol (Table 3).

### 2.3. Molecular Dynamics

MD studies were performed to simulate the dynamic trend and stability of the molecular Que-enzyme complexes obtained from the docking studies. In detail, the Desmond software (Schrödinger Release 2022-1) [39] was used to run all simulations for 500 ns by setting the Que-enzyme complex with the best Glide Gscore as the initial structure of each dynamic.

#### 2.3.1. NADP(H) Oxidase

To study the stability of MD between Que and NOX, we calculated the trajectory RMSD, which is plotted in Figure 7A. All the frame trajectories are aligned on the reference frame (frame 0), and the average RMSD for the protein structure is 1.515 ± 0.114 Å, while Que averages 8.854 ± 1.427 Å. Throughout the 500 ns simulation, the protein remains relatively stable without significant variations, while Que stabilizes after approximately 82 ns.

To obtain detailed information on the binding energy and its variations during the simulation, MMGBSA calculations were performed on every 10th frame (see Section 4), with an average of −56.421 ± 8.635 kcal/mol. The frame with the lowest value is 4190 (419 ns), with −73.563 kcal/mol, while the frame 80 (80 ns) has the highest value at −22.723 kcal/mol.

During the MD simulation, Que forms different interactions with the binding site of the enzyme, some of which are stable, while most are intermittent. The most stable ones include the H-bonds with Ser 115 (observed 74% of the simulation time), Cys 133 (54%), and Leu 241 (43%) (Figure 7B and Appendix A). The only interaction consistently observed in both docking and MD is an H-bond with Cys 242 (10%).

To gain deeper insights into the conformational changes of Que during the MD simulation, we performed trajectory clustering. The results of the five most populated clusters were submitted to MMGBSA calculations, and the results are shown in Table 4, while Appendix A illustrates the positions of the clusters within the binding site, comparing them with the best Glide Gscore docking pose. Notably, the clusters exhibit different positions compared to the poses extracted during docking.

#### 2.3.2. Xanthine Oxidase

The MD of XO and Que indicates that the protein increases steadily, reaching a peak after 150 ns of simulation before stabilizing (Figure 8A). In contrast, Que follows a stable trend until 330 ns, after which its RMSD begins to increase, forming a series of peaks and plateaus. The average RMSD for XO is 2.271 ± 0.261 Å, while Que exhibits an average RMSD of 7.627 ± 5.620 Å.

The average MMGBSA dG binding is −31.369 ± 6.173 kcal/mol. Frame 1 (0.1 ns) has the lowest value at −55.603 kcal/mol, while frame 3930 (393 ns) records the highest value at −10.938 kcal/mol.

The analysis of the interactions disclosed that Que forms stable interactions with Ala 1079 (H-bond, 40%), Glu 879 (water bridge, 38%), and Pro 1076 (water bridge, 33%) of NOX (Figure 8A) and maintains a H-bond with Glu 802 (21%), and two π-π stackings with Phe 1009 (16% and 10%), also present in the docking (Figure 8B).

We also clustered the trajectory script, and the results are provided in Table 5. The first cluster has an MMGBSA energy of −32.656 kcal/mol, while Appendix A depicts the alignment of the five clusters with the docking pose. In this case, the docking position and the clusters are positioned in the same portion, but the five clusters are like each other in the binding while the docking position differs (Appendix A).

#### 2.3.3. Myeloperoxidase

The average RMSDs for the protein structures of MPO and Que are 1.352 ± 0.098 Å and 9.462 ± 3.813 Å, respectively. While the RMSD of MPO is generally stable throughout the simulation, that of Que tends to increase until it stabilizes after 214 ns (Figure 9A).

Subsequently, MMGBSA was calculated in frames with an average of −30.663 ± 7.097 kcal/mol, and the lowest value is observed at frame 2710 (271 ns) with −62.350 kcal/mol, while the highest is recorded at frame 160 (16 ns) with −14.895 kcal/mol. In this case, the results align with the MMGBSA dG binding values obtained from the docking pose.

Analysis of the binding interactions between Que and MPO showed that the only interaction between Que and the active site residues for at least 10% of the simulation time is an H-bond with Glu 116 (19%), which can also be observed in the docking simulation (Figure 9B).

Finally, all frames of the simulation were clustered, and the top five clusters were reported in Table 6 with the MMGBSA energy values that were calculated. An analysis of the binding modes of the 5 clusters shows that Que occupies MPO binding sites differently (Appendix A). Clusters 2, 4, and 5 are located within the binding site, effectively replacing the space previously occupied by the docking pose, while clusters 1 and 3 gradually move out of the binding site.

#### 2.3.4. Lipoxygenase

The RMSD analysis was performed, obtaining an average RMSD of 1.557 ± 0.103 Å for 5-LOX, while that of Que was 2.881 ± 0.441 Å. The protein’s RMSD trend is stable, while that of Que is quite stable throughout the simulation but shows a series of peaks at intervals of 29–35 ns, 175–210 ns, and 377–380 ns (Figure 10A).

Then, the MMGBSA calculations performed on the trajectory yielded an average MMGBSA dG binding of −30.663 ± 7.097 kcal/mol, ranging from −51.253 kcal/mol at frame 20 (2 ns) to −21.253 kcal/mol at frame 1910 (191 ns). These results do not deviate significantly from those obtained from the MMGBSA calculations performed on the docking poses.

The analysis of interactions between Que and the 5-LOX showed that the most stable interactions are two π-cation interactions with Arg 101 (69% and 35%), also found in docking studies, and a water bridge with Tyr 142 (41%). Conversely, while the interactions with residues Glu 134, Arg 138, and Asp 166 establish a series of H-bonds during the docking simulation, in the MD, all these interactions transform into water bridges (32%, 18%, and 16%, respectively) (Figure 10B).

In addition, a cluster analysis on all frames of the simulation pointed out that the five most populated clusters are all located in a region of the binding site also occupied by the reference docking pose, but the binding mode differs slightly (Appendix A). Table 7 shows the results of the MMGBSA energy calculated for the five most populated clusters.

#### 2.3.5. Monoamine Oxidase

The RMSD analysis shows high RMSD values for the protein compared to previous simulations. The RMSD of MAO-A, with an average of 5.430 ± 1.001 Å, displays instability with a high peak of approximately 180 ns, while in the final part, approximately 400 ns, it finds stability. The RMSD of Que is unstable in the early part of the simulation and finds its stability after approximately 179 ns, and its average is 3.039 ± 0.562 Å (Figure 11A).

The MMGBSA calculations executed on the trajectory displayed an average MMGBSA of −38.836 ± 4.158 kcal/mol, with a minimum of (−53.135 kcal/mol) and a maximum of (−26.992 kcal/mol). Such results exhibit a significant discrepancy compared to the MMGBSA dG binding value of the docking pose, in which it even assumes a positive value (see Table 3).

The analysis of Que binding with MAO-A showed that the most frequent bonds are two H-bonds with Thr 336 (96%) and Cys 323 (79%), a water bridge with Asn 181 (74%), and a π-π stacking interaction with Phe 208 (68%) also present in the docking (Figure 11B and Appendix A).

In addition, cluster analysis showed that the five most populated clusters are all located in a region of the binding site also occupied by the reference docking pose (Appendix A), and the MMGBSA energies are shown in Table 8.

## 3. Discussion

Que is a potent anti-oxidant in the flavonoid family due to the presence of a phenolic hydroxyl group and double bonds [40].

In recent years, several in silico studies have been conducted to elucidate the molecular basis of its anti-oxidant activity.

Although the different DFT studies cannot be directly compared due to the use of different software and levels of theory, the HOMO and LUMO orbitals carried out by us and Zheng et al. [34] presented similar results, with the HOMO orbitals predominantly located in the B and C ring and the LUMO orbitals distributed throughout the molecule. Reported values are good indicators of chemical reactivity, especially for aromatic systems [35,36], and the results suggest that Que is a highly effective free radical scavenger.

Que anti-oxidant effects may, in part, arise from its ability to inhibit these enzymes. In fact, it has been reported that Que’s ability to attenuate the expression and enzymatic activity of various endogenous oxidative enzymes, including NOX, XO, LOX, MPO, and MAO, both in vivo and in vitro [9,10,11]. To validate these observations, we conducted docking analyses and molecular dynamics simulations.

Regarding the NOX enzyme, the results of our molecular docking study raise interesting questions regarding the inhibitory potential of Que on NOX activity, especially for the MMGBSA result (−4.82 kcal/mol). These results warrant further investigation into the dynamics and binding kinetics of Que within the NOX binding pocket. The MD confirms that the binding between Que and NOX finds excellent stability after approximately 100 ns and remains stable throughout the remainder of the simulation. The new MMGBSA calculations also improved by obtaining an average of −56.421 kcal/mol. The difference is probably due to the change in binding compared to docking and the better fit of Que on NOX observed during the simulation, which is evident from the greater formation of strong interactions for most of the simulation (Appendix A).

The previous studies, including X-ray structural analysis [37] and various docking simulations [38,41], consistently highlighted Que’s potential as an XO inhibitor. Building upon these promising insights, our own docking studies reaffirmed the previous findings supporting Que’s potential as an XO inhibitor. Moreover, MMGBSA energy in the docking pose yielded a substantial binding energy of −22.18 kcal/mol, indicating a discrete binding affinity confirmed by MD. The five most populated clusters also maintained binding in the same docking zone, and Que maintained excellent stability and some interactions for most of the simulation (Appendix A), losing it slightly but then regaining it.

Regarding MPO, our results reaffirm the potential inhibitory role of Que in modulating MPO activity, as previously studied by Pereira and colleagues [42]. Que effectively interacts with MPO, primarily through H bonds with Glu 116. The Glide Gscore energy values were not optimal, while the MMGBSA of the docking pose yielded a moderate energy result. The MD study revealed changes in Que binding within the binding site, as evidenced by the ligand’s RMSD exhibiting a wide range, especially at the beginning of the dynamics, where Que undergoes modifications in its binding. This is further confirmed by the range of MMGBSA values obtained, ranging from −62.350 to −14.895 kcal/mol, and by the different binding poses found in the five most populated clusters.

Regarding lipoxygenase (5-LOX) enzymes, our results are like the docking results presented by Vyshnevska et al. [43]. The MD studies confirm the binding stability between Que and this enzyme with low RMSDs for enzyme and ligand, and the MMGBSA studies performed throughout the dynamics are comparable to docking. Furthermore, the five most populated clusters have bindings comparable to docking. These results contribute to the growing body of evidence supporting Que’s role as a potential natural agent for controlling inflammation and related disorders through 5-LOX inhibition.

Comparing our results to those reported by Larit et al. [44], we observed some differences in Que’s interactions with MAO-A. Larit et al. reported that Que interacted with multiple residues, including Ala 111, Ile 180, Asn 181, Phe 208, Gln 215, Thr 336, and Tyr 444. In contrast, our docking study on the MAO-A active site showed that Que primarily formed an H-bond with Ala 111 and a π-π interaction with Phe 208. An anomalous result was found for the energy of MMGBSA for docking. MD studies overturned this positive value and obtained an average MMGBSA energy of −38.836 kcal/mol. Dynamics analysis showed that Que stabilizes after approximately 180 ns of simulation and then maintains its stability. The interactions with Ala 111 and Phe 208 are also preserved during MD, additionally forming those with Thr 336, Asn181, Cys 323, and Tyr 444. Overall, these results suggest that Que has the potential to inhibit MAO-A, albeit with variations in the specific residues involved.

## 4. Materials and Methods

### 4.1. DFT Approach

Jaguar (Schrödinger Release 2022-1: Jaguar, Schrödinger, LLC, New York, NY, USA, 2022) [45] was utilized to conduct the DFT calculations [28] using the B3LYP (Becke’s three-parameter exchange potential and Lee-Yang-Parr correlation function) method [46] in conjunction with the 6-311++G** (d, p) basis set, which successfully recommended itself for similar tasks and objects [47,48]. The calculations included the determination of the following properties: HOMO, LUMO, their energy gap (ΔE), ESP, ALIE, NBO, and ED.

### 4.2. Protein Preparation

On the Protein Data Bank (RCSB PDB) [49], we selected 5 X-ray structures, which were prepared using the Protein Preparation Wizard tool (Schrödinger Release 2022-1: Protein Preparation Wizard) [50]. The preparation included the removal of crystallized molecules, filling in missing side chains and loops, determining bond orders using the Chemical Component Dictionary (CCD) database [51], and adding hydrogen atoms. Epik (Epik, Schrödinger, LLC, New York, NY, USA, 2021) was employed to generate the states of heteroatoms at pH 7.4 ± 2.0. Protein optimization utilized PROPKA [52], and heavy atoms were converged to achieve an RMSD of 0.30 Å.

### 4.3. Ligand Preparation

Conformational analysis of Que was conducted using the Quantomechanic (QM) Conformer and Tautomer Prediction tool within Jaguar. The most favorable conformation was subsequently prepared using LigPrep (Schrödinger Release 2022-1: LigPrep, Schrödinger, LLC, New York, NY, USA, 2022) with the OPLS4 force field. and pH range of 7.0 ± 2.0 while preserving the specified chirality.

### 4.4. Molecular Docking

Among various docking programs [53,54], Glide (Schrödinger Release 2022-1: Glide, Schrödinger, LLC, New York, NY, USA, 2022) [55,56] was selected for the docking simulations, and three poses per ligand were retained using a threshold of 0.50 kcal/mol, and the OPLS_2005 [57] force field was employed.

To create grids, the Receptor Grid Generator was utilized. The center of the grid, with dimensions of 46 × 46 × 46 Å, was determined based on the centroid of the residues within the binding site. 5 grids were generated, one for each enzyme.

The docking results were subjected to binding energy calculation with MMGBSA (Molecular Mechanics with Generalized Born and Surface Area solvation) [58] of Prime (Schrödinger Release 2022-1: Prime, Schrödinger, LLC, New York, NY, USA, 2022) [59,60] using VSGB [61] as the solvation model, and OPLS4 [62] as the force field.

### 4.5. Molecular Dynamics

MD simulations were conducted using Desmond (Schrödinger Release 2022-1: Desmond Molecular Dynamics System, New York, NY, USA, 2022 [39]. For each system, the pose with the best docking score was chosen, and a system setup was performed. The TIP3P (Transferable Intermolecular Potential with 3 Points) [63] solvent model was used, and the ligand-receptor complex was placed in an orthorhombic water box that extended 10.0 Å, and the box volumes were minimized and neutralized by adding ions (Na^+^ or Cl^−^). The OPLS4 force field was selected.

MD simulations were carried out for 500 ns, with trajectory recorded at 100 ps, in the NPT ensemble, maintaining a constant temperature (300.0 K) and pressure (1.01325 bar) using the Nosé-Hoover thermostat [64] and the Martyna-Tobias-Klein barostat [65] methods, respectively. Other parameters were kept at their default settings. The behavior and interactions between Que and the proteins were analyzed by utilizing the Simulation Interaction Diagram tool integrated into Desmond [39].

The trajectories obtained were submitted to a post-MMGBSA analysis using the thermal_mmgbsa.py script implemented in Desmond [39]. Out of 5001 frames, one frame was analyzed every 10 ns, resulting in a total of 500 frames analyzed.

Additionally, trajectory frames were clustered based on Que’s RMSD using Schrödinger’s trj_cluster.py script [66], a fully automated clustering procedure, and output files were generated for the 5 most populated clusters in each MD experiment. Each cluster was then submitted to MMGBSA calculations with Prime [59,60].

## 5. Conclusions

Que, a flavonoid widely distributed in fruits and vegetables, is known for its anti-microbial, anti-viral, anti-oxidant, and anti-inflammatory properties. Its anti-oxidant activity is mainly attributed to its intrinsic ability to neutralize free radicals due to the hydroxyl substitutions and the catechol-type B-ring. Additionally, it can inhibit the expression of pro-inflammatory and oxidant genes and endogenous oxidizing enzymes. The results of docking studies show that Que can interact with oxidative enzymes, making it a promising candidate for developing new anti-inflammatory agents. The study found that Que demonstrated the greatest effectiveness in molecular docking simulations with XO, followed by MAO-A, 5-LOX, NOX, and MPO, which are implicated in several inflammatory pathologies. To confirm these findings and explore the mechanisms of action further, MD simulation studies were conducted over a 500 ns period. These simulations confirmed binding and contributed to a more detailed understanding of the interactions between Que and the five oxidative enzymes. For the NOX and MPO simulations, clear variations in binding with Que were observed, while excellent binding energy values calculated with MMGBSA were observed for all simulations. In addition, all enzymes except MPO showed strong interactions preserved during the simulations. In fact, the only stable and weak interaction with MPO was with Glu116 (19%). These findings collectively emphasize the remarkable potential of Que as a versatile compound with significant implications for its development as a novel class of anti-inflammatory agents, provided strategies to enhance its bioavailability are explored and implemented.

However, it is crucial to acknowledge that Que has its limitations, particularly in terms of bioavailability. Its aglycone form exhibits low systemic exposure. This is an important consideration for the potential therapeutic use of Que. Novel delivery methods, including inclusion complexes, prodrugs, nanocrystals, microemulsions, liposomes, and phospholipid formulations, as well as polymer nanoparticles and micelles, hold significant promise and warrant further exploration to improve que’s bioavailability, medicinal value, and applicability.

## Figures and Tables

**Figure 1 ijms-24-15391-f001:**
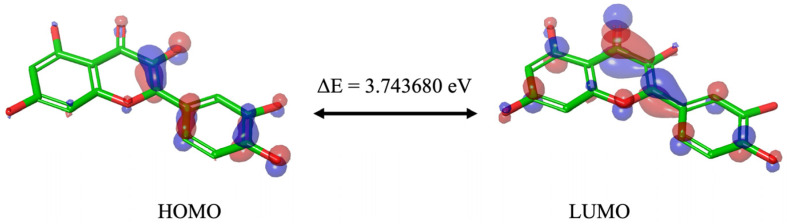
Molecular orbitals of Que and energy gap (ΔE). Blue indicates a positive phase, while red corresponds to a negative phase.

**Figure 2 ijms-24-15391-f002:**
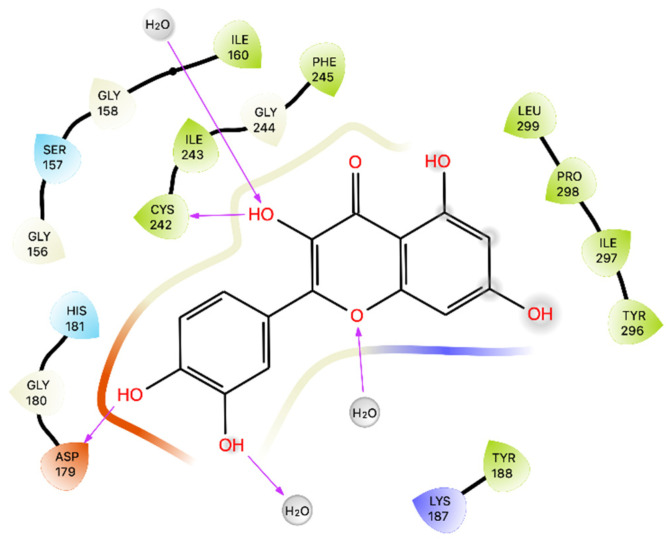
Ligand interaction diagram of Que docked in NOX. Purple arrows represent H-bonds.

**Figure 3 ijms-24-15391-f003:**
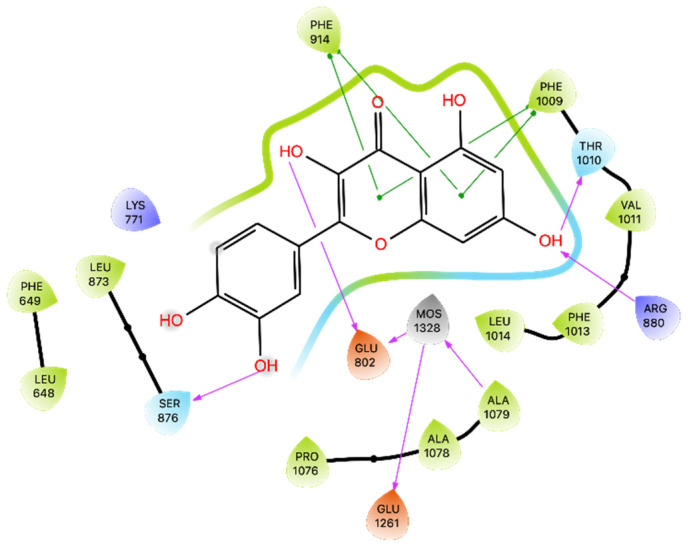
Ligand interaction diagram of Que docked in XO. Purple arrows represent H-bonds, whereas green lines represent π-π stackings. The MOS 1328 is the dioxothiomolybdenum (VI) ion.

**Figure 4 ijms-24-15391-f004:**
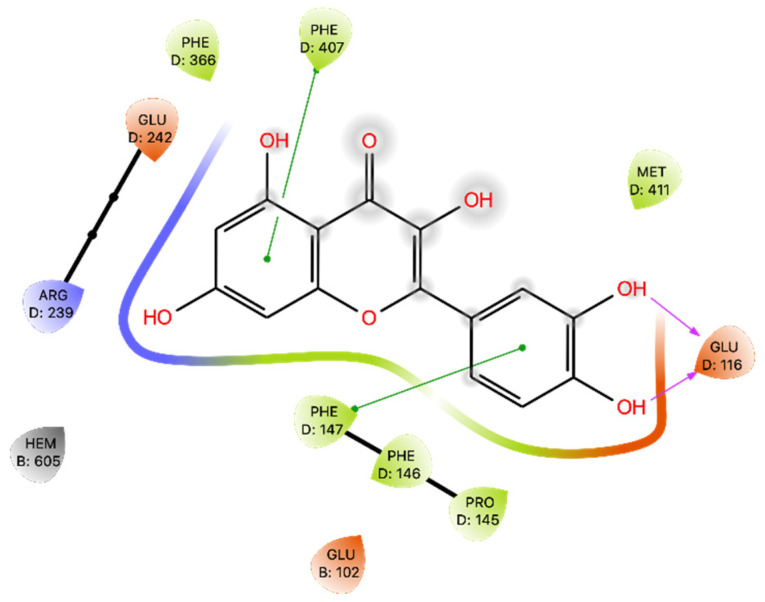
Ligand interaction diagram of Que docked in MPO. Purple arrows represent H-bonds, whereas green lines represent π-π stackings.

**Figure 5 ijms-24-15391-f005:**
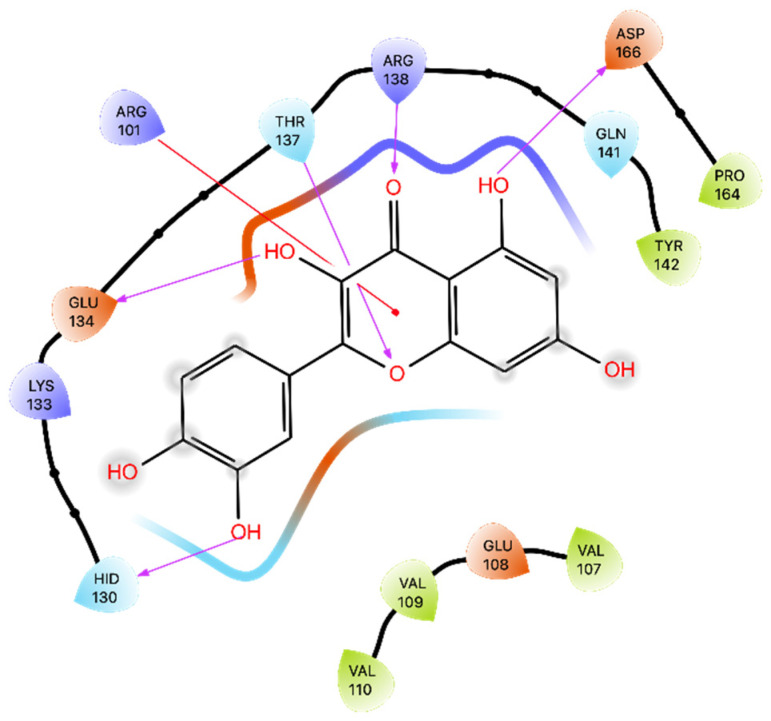
Ligand interaction diagram of Que docked in 5-LOX. Purple arrows represent H-bonds, and red lines represent π-cation interactions.

**Figure 6 ijms-24-15391-f006:**
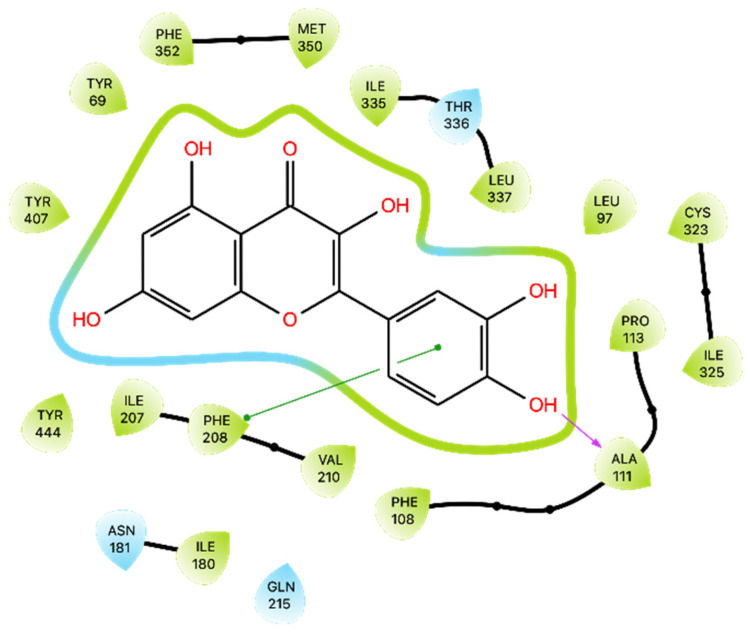
Ligand interaction diagram of Que docked in MAO-A. Purple arrows represent H-bonds, whereas green lines represent π-π stackings.

**Figure 7 ijms-24-15391-f007:**
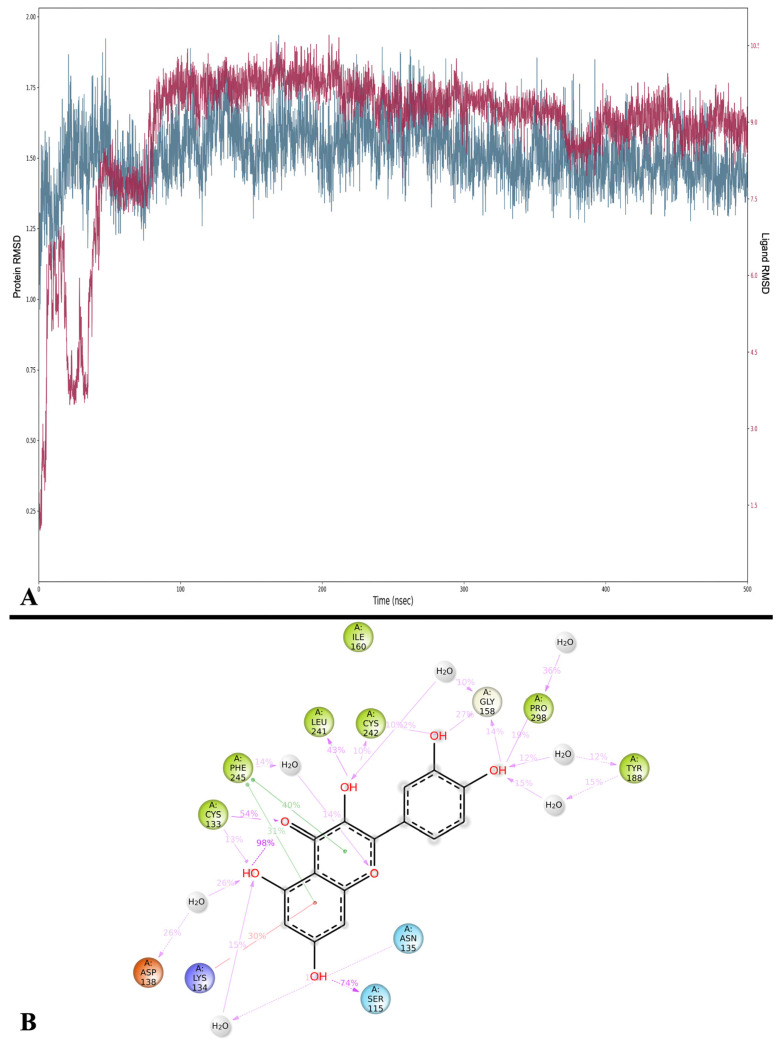
(**A**) RMSD analysis of Que (Lig fit on Prot, red) and NOX (Cα, blue). On the *x*-axis, the simulation time is represented in ns, while on the *y*-axis, the protein RMSD (left) and ligand RMSD (right) are plotted. (**B**) Detailed interactions between Que and NOX. Interactions happening for more than 10% of the simulation time are shown. Purple arrows represent H-bonds, while green lines represent π-π stackings.

**Figure 8 ijms-24-15391-f008:**
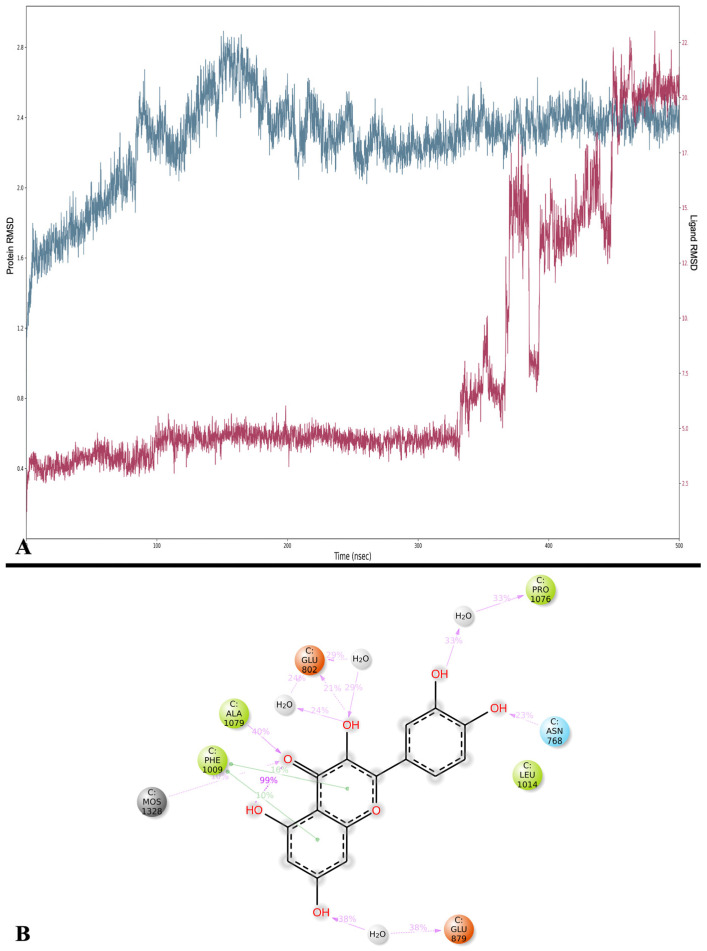
(**A**) RMSD analysis of Que (Lig fit on Prot, red) and XO (Cα, blue). On the *x*-axis, the simulation time is represented in ns, while on the *y*-axis, the protein RMSD (left) and ligand RMSD (right) are plotted. (**B**) Detailed interactions between Que and XO. Interactions happening for more than 10% of the simulation time are shown. Purple arrows represent H-bonds, while green lines represent π-π stackings.

**Figure 9 ijms-24-15391-f009:**
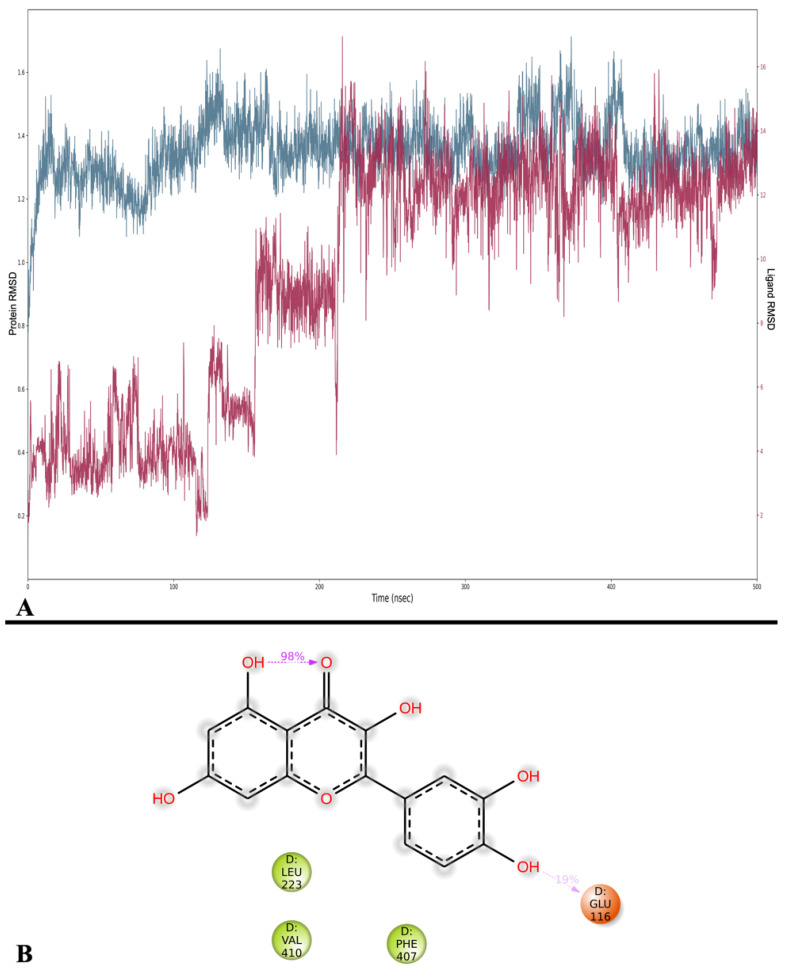
(**A**) RMSD analysis of Que (Lig fit on Prot, red) and MPO (Cα, blue). On the *x*-axis, the simulation time is represented in ns, while on the *y*-axis, the protein RMSD (left) and ligand RMSD (right) are plotted. (**B**) Detailed interactions between Que and MPO. Interactions happening for more than 10% of the simulation time are shown. Purple arrows represent H-bonds.

**Figure 10 ijms-24-15391-f010:**
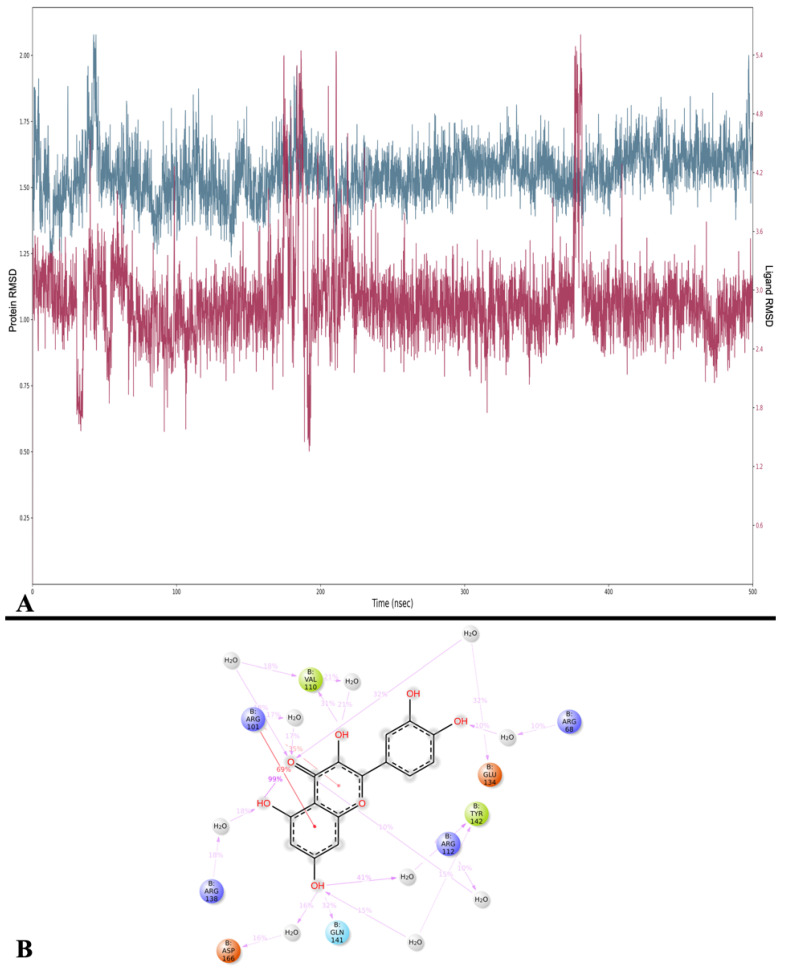
(**A**) RMSD analysis of Que (Lig fit on Prot, red) and 5-LOX (Cα, blue). On the *x*-axis, the simulation time is represented in ns, while on the *y*-axis, the protein RMSD (left) and ligand RMSD (right) are plotted. (**B**) Detailed interactions between Que and 5-LOX. Interactions happening for more than 10% of the simulation time are shown. Purple arrows represent H-bonds, while red lines represent π-cation interactions.

**Figure 11 ijms-24-15391-f011:**
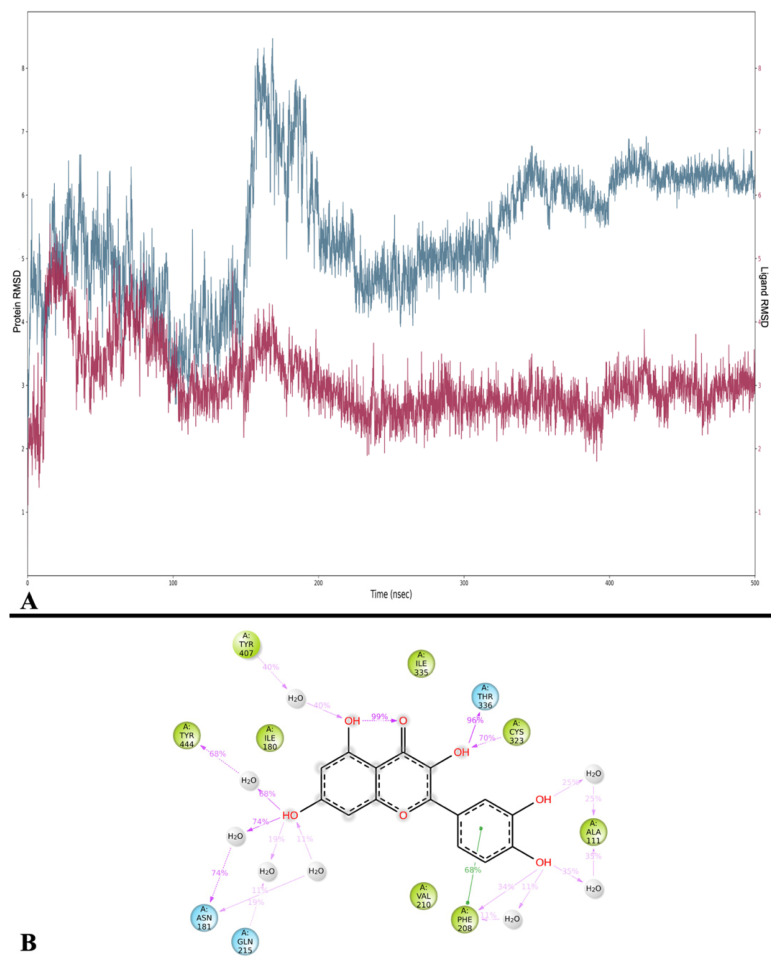
(**A**) RMSD analysis of Que (Lig fit on Prot, red) and MAO-A (Cα, blue). On the *x*-axis, the simulation time is represented in ns, while on the *y*-axis, the protein RMSD (left) and ligand RMSD (right) are plotted. (**B**) Detailed interactions between Que and MAO-A. Interactions happening for more than 10% of the simulation time are shown. Purple arrows represent H-bonds, while green lines represent π-π stackings.

**Table 1 ijms-24-15391-t001:** Theoretical results of Que using B3LYP/6-311++G** method with Jaguar software (Schrödinger Release 2022-1).

**HOMO**	−5.68 eV
**LUMO**	−1.94 eV
**ΔE HOMO-LUMO**	3.74 eV
**IP**	5.68 eV
**EA**	1.94 eV
**ESP**	−2.65 eV (min value) 3.18 eV (max value)
**ALIE**	8.84 eV (min value) 16.81 eV (max value)

**Table 2 ijms-24-15391-t002:** Selected X-ray structures from the Protein Data Bank.

PDB ID	Resolution (Å)	Enzyme	Organism	Co-Crystallized Ligand(s)	Ref.
2CDU	1.80	NOX	Fructilactobacillus sanfranciscensis	Flavin-adenine dinucleotide (FAD), adenosine-5′-diphosphate (ADP)	[34]
2Z5Y	2.17	MAO-A	Homo sapiens	Flavin-adenine dinucleotide (FAD), decyl(dimethyl)phosphine oxide (DCX), 7-methoxy-1-methyl-9h-beta-carboline (HRM)	[35]
3NVY	2.00	XO	Bos taurus	Molybdopterin (MTE), quercetin (QUE)	[36]
5FIW	1.70	MPO	Homo sapiens	2-acetamido-2-deoxy-beta-D-glucopyranose (NAG), beta-D-mannopyranose (BMA)	[37]
6NCF	2.87	5-LOX	Homo sapiens	(3alpha,8alpha,17alpha,18alpha)-3-(acetyloxy)-11-oxours-12-en-23-oic acid (AFX)	[38]

**Table 3 ijms-24-15391-t003:** Docking and Prime MMGBSA results for the interactions of Que with the analyzed enzymes. The results are all expressed in kcal/mol.

Enzyme	Glide Gscore	MMGBSA dG Bind
NOX	−5.35	−4.82
XO	−8.87	−22.18
MPO	−4.91	−28.81
5-LOX	−5.94	−37.42
MAO-A	−8.14	5.72

**Table 4 ijms-24-15391-t004:** Calculated clusters derived from the trajectory of the MD simulation on the Que-NOX complex.

Cluster	N. of Members	MMGBSA dG Bind
1	62	−47.260 kcal/mol
2	59	−48.970 kcal/mol
3	57	−55.649 kcal/mol
4	56	−51.466 kcal/mol
5	56	−54.908 kcal/mol

**Table 5 ijms-24-15391-t005:** Calculated clusters derived from the trajectory of MD simulation on the Que-XO complex.

Cluster	N. of Members	MMGBSA dG Bind
1	151	−32.656 kcal/mol
2	97	−39.741 kcal/mol
3	96	−36.597 kcal/mol
4	91	−37.437 kcal/mol
5	86	−28.251 kcal/mol

**Table 6 ijms-24-15391-t006:** Calculated clusters derived from the trajectory of MD simulation on the Que-MPO complex.

Cluster	N. of Members	MMGBSA dG Bind
1	108	−30.793 kcal/mol
2	96	−22.584 kcal/mol
3	94	−23.547 kcal/mol
4	89	−33.276 kcal/mol
5	82	−29.198 kcal/mol

**Table 7 ijms-24-15391-t007:** Calculated clusters derived from the trajectory of MD simulation on the Que-5-LOX complex.

Cluster	N. of Members	MMGBSA dG Bind
1	58	−40.127 kcal/mol
2	55	−41.856 kcal/mol
3	50	−43.550 kcal/mol
4	50	−35.036 kcal/mol
5	48	−36.262 kcal/mol

**Table 8 ijms-24-15391-t008:** Calculated clusters derived from the trajectory of MD simulation on the Que-MAO-A complex.

Cluster	N. of Members	MMGBSA dG Bind
1	156	−43.672 kcal/mol
2	154	−40.106 kcal/mol
3	146	−29.574 kcal/mol
4	137	−37.437 kcal/mol
5	131	−34.486 kcal/mol

## Data Availability

Not applicable.

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
