# Peer review of "Inhibitory Effect of Quercetin on Oxidative Endogen Enzymes: A Focus on Putative Binding Modes"

_ijms, 2023, doi:10.3390/ijms242015391_

Round 1

Reviewer 1 Report

1. Overview:

The manuscript presents a detailed exploration of Quercetin's (Que) inhibitory effects on oxidative endogenous enzymes. While the study is methodically sound and well-structured, there are areas that necessitate refinement to bolster the manuscript's clarity, depth, and overall presentation.

2. Major Points for Consideration:

Figures' Clarity and Composition: Figures 8 to 12 suffer from resolution issues, with text and numerical data being too diminutive for clear interpretation. It is imperative to enhance the resolution and ensure legibility. Additionally, Figure 1 appears overly simplistic and might be redundant. Consideration should be given to consolidating certain figures into composite figures with multiple subplots for a more streamlined presentation.

Content Refinement: The introduction and results sections are protracted and would benefit from succinctness. A focused approach, emphasizing pivotal points and minimizing repetition, would enhance the manuscript's readability.

Methodological Justification: The manuscript employs a variety of computational techniques. A more explicit elucidation regarding the choice of specific methods and their synergistic relevance would provide readers with a clearer understanding of the study's approach.

Discussion Structure: The discussion, while thorough, could be more organized. Categorizing findings based on individual enzymes and subsequently delving into broader implications would make the section more reader-friendly.

Addressing Limitations: The limitations of Que, particularly its bioavailability, are briefly mentioned. A more comprehensive discussion on how these limitations might influence the study's conclusions and their practical applications is warranted.

3. Minor Points for Consideration:

Clarity in "Materials and Methods": Certain segments, especially within the "Materials and Methods," require rewording to enhance clarity and ensure a seamless flow of information.

Comparative Analysis: A deeper comparative analysis of Que's interactions across different enzymes would be beneficial. Emphasizing the most potent interactions and providing a rationale for the observed effects would enrich the study's depth.

Future Research Directions: The conclusion should more explicitly outline avenues for future research, particularly strategies to augment Que's bioavailability and address its limitations.

4. Conclusion:

The manuscript provides insightful revelations into Que's interactions with oxidative endogenous enzymes. To optimize the presentation and understanding of the study's findings, considerable revisions are recommended, particularly in terms of figure clarity, content organization, and depth of discussion.

While the manuscript is generally well-written, there are sections, especially in the "Conclusion," where sentences could benefit from minor rephrasing for clarity.

Author Response

Response to Reviewer 1 comments

We thank you for your review. Your feedback helped us to improve and clarify some parts of our manuscript. We are truly grateful for the time and effort you put into this process.

  1. Major Points for Consideration:

Figures' Clarity and Composition: Figures 8 to 12 suffer from resolution issues, with text and numerical data being too diminutive for clear interpretation. It is imperative to enhance the resolution and ensure legibility. Additionally, Figure 1 appears overly simplistic and might be redundant. Consideration should be given to consolidating certain figures into composite figures with multiple subplots for a more streamlined presentation. 

Thank you for your observation. We have changed and improved the resolution of Figures 8 to 12, which are now Figures 7 to 11 and we have removed Figure 1 as suggested. Unfortunately, the figures for the results of the dynamics are derived from the software and do not allow the resolution to be changed. We have however improved them. Regarding the suggestion of merging the figures, the guidelines in the IJMS submission form recommend inserting figures immediately after they are cited. Merging the docking figures would entail placing the combined image within only one of the subsections, potentially making the results of each docking less visually distinct and understandable.

Content Refinement: The introduction and results sections are protracted and would benefit from succinctness. A focused approach, emphasizing pivotal points and minimizing repetition, would enhance the manuscript's readability.

Thank you for your remark. We have shortened and simplified the introduction and results to make them more fluent.

Methodological Justification: The manuscript employs a variety of computational techniques. A more explicit elucidation regarding the choice of specific methods and their synergistic relevance would provide readers with a clearer understanding of the study's approach.

Thank you for your observation. To better clarify our methodology, we have added sentences (150-156 of marked copy version) in the introduction.

Discussion Structure: The discussion, while thorough, could be more organized. Categorizing findings based on individual enzymes and subsequently delving into broader implications would make the section more reader-friendly.

Thank you for your point. We have edited the discussion by simplifying it and trying to make it more reader-friendly

Addressing Limitations: The limitations of Que, particularly its bioavailability, are briefly mentioned. A more comprehensive discussion on how these limitations might influence the study's conclusions and their practical applications is warranted.

Thank you for your observation. We have added the limitation of quercetin's bioavailability both in the introduction and in the final conclusions.

  1. Minor Points for Consideration:

Clarity in "Materials and Methods": Certain segments, especially within the "Materials and Methods," require rewording to enhance clarity and ensure a seamless flow of information.

Thank you for your suggestion. We have slimmed down and simplified the “Materials and Methods”

Comparative Analysis: A deeper comparative analysis of Que's interactions across different enzymes would be beneficial. Emphasizing the most potent interactions and providing a rationale for the observed effects would enrich the study's depth.

Thank you for your observation. We have included in the supplementary a Table of Que's interaction with the 5 enzymes during molecular dynamics (Table S4) that we mentioned in the results, and added in the discussion.

Future Research Directions: The conclusion should more explicitly outline avenues for future research, particularly strategies to augment Que's bioavailability and address its limitations.

Thank you for your observation. We have included this aspect in the conclusions.

  1. Conclusion:

The manuscript provides insightful revelations into Que's interactions with oxidative endogenous enzymes. To optimize the presentation and understanding of the study's findings, considerable revisions are recommended, particularly in terms of figure clarity, content organization, and depth of discussion.

Thank you for the suggestions. We have accordingly modified.

Comments on the Quality of English Language

While the manuscript is generally well-written, there are sections, especially in the "Conclusion," where sentences could benefit from minor rephrasing for clarity.

Thank you for the suggestions. We have accordingly modified.

Reviewer 2 Report

PDF file attached

Minor editing of the English language is needed to enhance the overall quality of the manuscript

Author Response

Response to Reviewer 2 comments

Thank you for your review of our work. Your feedback helped us to improve and clarify some parts of our work. We are truly grateful for the time and effort you have put into this process.

1. It would be beneficial if the authors explicitly justified their choice of the B3LYP/6-311++G** basis set functional. Explaining why this specific basis set was chosen over others and how it aligns with the objectives of the study would provide clarity to the readers. 

Thank you for pointing this out. The choice of B3LYP/6-311++G** for our DFT study of quercetin is motivated by its suitability for handling the electronic structure of quercetin and its proven accuracy in describing similar molecular systems. We agree with this comment and have therefore added the following sentence in the text “which successfully recommended itself for the similar tasks and objects” (Materials and Methods line 600-601 of marked copy version) and added references 57 and 58.

2. The authors could elaborate on the accuracy and reliability of the B3LYP/6-311++G** basis set functional in predicting the electronic structure and reactivity properties of the molecules under investigation.

Thank you for your input, we have included the structure in the supplementary Figure S1A and inserted the following sentence in the results: “Density Functional Theoretical (DFT) approaches at the B3LYP/6-311++G(d, p) level was applied for the optimisation of Que, and reveals the presence of C2′H/C3OH, C3OH C4O, C4O/C5O H bonds (Figure S1A)”(line 160-162 of marked copy version).

3. It is highly recommended that the authors provide the coordinates of all the systems studied in the manuscript. The inclusion of molecular coordinates is essential for enhancing the reproducibility and transparency of the study. Providing these coordinates as supporting information or depositing them in a publicly accessible repository such as Zenodo would greatly benefit the scientific community.

Thanks for the suggestion, we inserted the calculated molecular coordinates of quercetin derived from our DFT studies in the supplementary file (Table S1A). We also included bond lengths (Table S1B), bond angles (Table S1C), and torsional angles (Table S1D). Regarding the coordinates of all the studied systems, we will consider uploading the docking results and the molecular dynamics trajectories to Zenodo or other database once the article is published.

4. It is recommended that the authors utilize an alternative docking tool, EnzyDock (J. Chem. Theory Comput. 2019, 15, 9, 5116), which was recently published. Furthermore, it is essential to properly cite EnzyDock in the manuscript.

Thank you for the advice. We are not familiar with this tool, but we have downloaded it and will certainly test it for use in our upcoming studies. In our work, we have relied on a pipeline entirely based on the Schrödinger suite (Jaguar, Glide, Prime, Desmond, etc.), and introducing a recent docking tool that we have never used does not align with our workflow, although we will mention it as a potential tool to use in materials and methods subsection molecular docking. However, we warmly welcome this recommendation for our future studies.

5. The authors mentioned utilizing a Python script for trajectory clustering. it’s recommended to provide this script in GitHub. Additionally, specifying the clustering methodology employed, and whether it was a fully automated process, would aid in understanding the data analysis approach. This information is crucial for readers to replicate or adapt the clustering method.

Thank you for your recommendation. We have added an explanation regarding the automation of the trajectory clustering script: “a fully automated clustering procedure” (line 670, paragraph 4.5 Molecular Dynamics. However, it is not possible to upload it on GitHub. This script is included within the Script Center (https://www.schrodinger.com/scriptcenter?f%5B0%5D=field_command_line_script_type%3A1119) and is available to those who own Schrödinger licenses.

6. In analyzing the HOMO and LUMO of the system, incorporating Natural Bond Orbital (NBO) analysis can provide deeper insights. NBO analysis offers a more detailed understanding of electronic structures and bonding, complementing the HOMO and LUMO assessments. Integrating NBO analysis could enhance the robustness and thoroughness of the study’s electronic structure characterization.

Thank you very much, as you suggested we have also included the NBO analysis. We have included the results of the “Natural population analysis” in Table S2 and “Second order perturbation analysis of Fock matrix” in Table S3 of the supplementary files.

Upon addressing these revisions, it is believed that the manuscript will be suitable for publication. Overall appreciation is expressed for the study’s findings, particularly the comprehensive conclusions regarding Que’s potential as an anti-inflammatory agent. The manuscript provides valuable insights into the mechanisms of action underlying Que’s antioxidant and anti-inflammatory properties. In conclusion, the authors are encouraged to address the minor revisions outlined above. Once these concerns are taken into account, it is believed that the manuscript will be well-suited for publication. The potential of the authors’ work to make a valuable contribution to the field of oxidative stress and natural therapeutics is acknowledged.